# In Silico Repurposed Drugs against Monkeypox Virus

**DOI:** 10.3390/molecules27165277

**Published:** 2022-08-18

**Authors:** Hilbert Yuen In Lam, Jia Sheng Guan, Yuguang Mu

**Affiliations:** 1School of Biological Sciences, Nanyang Technological University, 60 Nanyang Dr, Singapore 637551, Singapore; 2A*STAR Skin Research Labs, Agency of Science, Technology and Research, Singapore, 11 Mandalay Rd, #17-01, Singapore 308232, Singapore

**Keywords:** in silico, monkeypox, drug repurposing, epidemic, poxviruses

## Abstract

Monkeypox is an emerging epidemic of concern. The disease is caused by the monkeypox virus and an increasing global incidence with a 2022 outbreak that has spread to Europe amid the COVID-19 pandemic. The new outbreak is associated with novel, previously undiscovered mutations and variants. Currently, the US Food and Drug Administration (FDA) approved poxvirus treatment involves the use of tecovirimat. However, there is otherwise limited pharmacopoeia and research interest in monkeypox. In this study, virtual screening and molecular dynamics were employed to explore the potential repurposing of multiple drugs previously approved by the FDA or other jurisdictions for other applications. Several drugs are predicted to tightly bind to viral proteins, which are crucial in viral replication, including molecules which show high potential for binding the monkeypox D13L capsid protein, whose inhibition has previously been demonstrated to suppress viral replication.

## 1. Introduction

The monkeypox virus belongs to the clade *Orthopoxviruses*, which groups with smallpox, cowpox, vaccinia, and variola [1], and is endemic to West and Central Africa.

Previously, infections have been reported mainly in Central Africa, with the first case being diagnosed in the Democratic Republic of Congo (DRC) in 1970 [2]. A review by Bunge et al. (2022) indicated varying fatality rates ranging from 3.6 to 10.6% between different strains of monkeypox across Africa [2], hypothesising a growing potential for lethality. As of late, monkeypox cases have been found exported into Singapore [3], the United Kingdom [4], Israel [5], and the United States [6] by travellers. An outbreak in the Midwest region of the United States in 2003 was linked to prairie dog contact from infected rodents from Ghana [7]. None of these outbreaks had reported deaths due to monkeypox; however, the reported cases of these outbreaks ranged from a single case in Singapore, the United Kingdom, and Israel, to 71 cases in the United States in the 2003 outbreak.

The transmission of monkeypox occurs by an animal bite or direct contact with an animal’s bodily fluids [7]. It can also spread through respiratory droplets during direct and/or prolonged face-to-face contact, through direct contact with bodily fluids of an infected person, or via viral particle contaminated objects [7]. In the current outbreak, monkeypox is being debated as a sexually transmitted disease (STD) [8]. The prevention of monkeypox is 85% effective with the smallpox vaccine [9].

Monkeypox symptoms present as fever ranging between 38.5 °C and 40.5 °C, malaise, rash, and headaches. Most tellingly, hard and deep, well-circumscribed and umbilicated lesions are present with swelling of lymph nodes [1]. The monkeypox incubation period ranges from 7 to 17 days, with fever declining after 3 days of rash onset [1]. The lesions are noted to be swollen, stiff, and painful. Lymphadenopathy, which presents in monkeypox but not smallpox, has been hypothesised to elicit a stronger immune response, compared with smallpox [10]. Following the lesional segment, scabs are formed and eventually shed off, leading to discoloured spots [5]. Instances of sepsis due to lesions have been reported but are generally accepted to be rare [11].

Monkeypox is diagnosed through PCR, culture, immunohistochemistry such as ELISA, or electron microscopy, depending on test availability [5,9]. Viral particles from the case investigated by Erez et al. (2019) showed a particle size of 281 ± 18 nm × 220 ± 17 nm (*n* = 24). Treatment of monkeypox is supportive [9].

Currently identified potential treatments include tecovirimat, which is approved by the US FDA for the treatment of smallpox. Cidofovir or brincidofovir have also been listed as possibly effective for monkeypox treatment; however, their use has not been studied outside of experimental models [9]. The application of other drugs has not been widely explored, and therefore, an open knowledge gap exists in monkeypox treatment.

The monkeypox genome is similar to that of smallpox, with a previous study showing 96.3% identity in the central regions, which encodes essential enzymes and proteins [12]. Monkeypox genome nomenclature follows that of universal nomenclature but differs in certain regions which encode for virulence [13]. Out of all Poxviridae, vaccinia is one of the most well-studied due to its use in vaccination against other types of poxviruses [14] and being a much less dangerous relative of smallpox [15]. Monkeypox, smallpox, and vaccinia have a genome size of approximately 197 kb [16], 186 kb [17], and 190 kb, respectively [18]. All poxvirus genomes are linear, double-stranded DNAs.

For clarity in this work, due to the similarities and naming difficulties of other poxvirus homologues, the authors refer to the targets, homologues or otherwise, after vaccinia nomenclature as the universal nomenclature gene name as in vaccinia virus. Five poxvirus targets are investigated: (a) A48R, (b) A50R, (c) D13L protein trimer complex, (d) F13L, and (e) I7L, all of which have been proposed as useful intervention targets by other studies and reviews [19]. A50R, D13L, F13L, and I7L were chosen due to the previous literature supporting in vitro or in vivo evidence for their efficacy, whereas A48R has been previously identified to be a good potential drug target [19].

(a)A48R is a thymidylate kinase previously determined to complex with thymidine diphosphate [20] and is a novel target in the sense that it is currently not targeted by any known drugs. The function of A48R is to phosphorylate thymidine monophosphate to its diphosphate and also to phosphorylate 5′ halogenated deoxyuridine monophosphate analogues [19]. The substantial structural difference from human thymidylate kinase at the active site [20] makes it an attractive target for other thymidine analogues without excessive concern of interfering with the human analogue’s function.(b)A50R is a DNA ligase known to be inhibited by mitoxantrone. The N-terminal mutations of this protein demonstrated instances of antiviral resistance. The location of resistance mutations pinpoints its active site towards this location [21] and makes it a valuable target for drug discovery. Mitoxantrone has also been shown to be unique amongst the inhibitors identified, in that it has no impact on viral gene replication [21], but clear microscopic evidence shows that it blocks the assembly of viral particles by an unknown mechanism [21].(c)D13L is a protein trimer complex and a major capsid protein which lends rigidity to the membrane and is especially important in viral particle morphogenesis [22]. In previous studies, rifampin has been demonstrated to have an affinity for the D13L protein in the vaccinia virus [19]. Rifampin, synonym rifampicin, is an antibiotic produced by the soil bacteria *Amycolatopsis rifamycinica* and is used to treat tuberculosis, leprosy, and MRSA [23]. It has separately been demonstrated to inhibit poxvirus assembly by binding to the D13L trimer complex, a mechanism independent of its antibacterial activity [24]. Unfortunately, the rifampin effect of halting viral replication is reversible, with the assembly reinitiating within minutes after drug withdrawal [25]. Clinically, rifampin is not used due to the emergence of mutant, rifampin-resistant viruses [26], but with a larger pharmacopoeia, the authors propose that this problem may be overcome using drug cocktails. Despite the current lack of clinical use of rifampin in treating poxvirus infections, the D13L docking site was derived from X-ray crystallography in complex with rifampin [24], providing a targetable, active docking site for drug design. In previous studies, rifampin has exhibited the in vitro depression of vaccinia-directed protein synthesis and cowpox viral particles in HeLa-S_2_ cells [27]. The anti-poxviral activity of rifampin and the lengthening or shortening of chains have also previously been explored and found to affect specific antiviral activity [28].(d)F13L is the current target of the only approved drug for poxvirus treatment, tecovirimat (previously known as ST-246). It is a major envelope protein and is also a palmitoylated membrane protein [19], making it vital for the extracellular envelope virus (EEV) formation and important for viral entry into the cell [29]. Tecovirimat has been shown to be strongly efficacious in animal models [30].(e)I7L core proteinase is a cysteine proteinase which cleaves the major structural and membrane proteins of viruses [31]. Proteases are ideal therapeutic targets due to their essential function in viral replication by cleaving precursor polyproteins [32], and protease inhibitors in other viruses such as HIV have shown promise for other viral protease inhibitors [33]. As such, monkeypox virus proteases are also attractive targets to inhibit viral replication. Previous studies identified TTP-6171 as an inhibitor of I7L. However, drug-resistant I7L could also be generated with mutations [31].

After deriving these targets, the authors identified the active protein residues in monkeypox using multiple sequence alignments and, consequently, propose eight novel potentially repurposable drugs. These eight drugs are NMCT and rutaecarpine for A48R, nilotinib for A50R, simeprevir for D13L, hypericin and naldemedine for F13L, and fosdagrocorat and lixivaptan for I7L.

## 2. Results

### 2.1. Protein Similarity of Monkeypox Isolates and Chosen Docking Proteins

High protein similarity was observed throughout poxvirus targets in general (Figure 1). However, the French isolate showed premature stop codons inserted in all proteins. However, none of the active residues previously identified in the literature were mutated (Figure 1). Lower sequence conservation for I7L was observed near active site residue His-241 due to unclear sequencing of the Portugal isolate. The explorations of the deletions in the French isolate were performed away from any docking and active site (Appendix A).

### 2.2. Stability of Proteins

Proteins were shown to be stable (Appendix A), with little unexpected psi and phi angles throughout. Furthermore, the RMSD of the proteins to the initial structure was low (Appendix A), indicating that significant overall conformational changes throughout the simulation were absent. At a local level, the RMSF of residues remained low (mainly below 2 nm) throughout the entire simulation (Appendix A), showing no drastic local changes in conformational structure that could affect docking stability.

### 2.3. Active Sites and Protein Conformation

The sites of deletion were also facing away from the active site (Appendix A), indicating limited impact on experimentation. The direction away from the binding site indicates that the tested drugs worked across the different isolates tested in silico, as these sites were sufficiently distant from the target.

#### 2.3.1. Repurposable Drugs for A48R

NMCT and rutaecarpine were the drugs chosen (Table 1). The QVina2.1 binding affinity for NMCT and rutaecarpine are −7.0 kcal/mol and −9.0 kcal/mol, respectively. Thymidine diphosphate, the native ligand to A48R, has a docking energy value of −7.4 kcal/mol. NMCT was chosen due to its previously demonstrated activity in thymidine kinase of poxviruses [39] and a good pose in A48R (Figure 2). The RMSD of both NMCT and rutaecarpine remained low throughout the simulation (Figure 3), and both NMCT and rutaecarpine showed hydrophobic interactions (Figure 4). NMCT was also noted to form strong hydrogen bonds with some of the amino acid residues (Figure 5). Both NMCT and rutaecarpine were also clustered into one single stable cluster for the entire simulation (Table 2), further proving the ligand stability in the pocket.

#### 2.3.2. Repurposable Drugs for A50R

Nilotinib was the drug chosen (Table 1). It had the highest binding affinity of all the drugs screened, with a QVina2.1 score of −10.8 kcal/mol. The hydrophilic groups were also observed to be spatially close to the hydrophilic regions of the amino acid (Figure 2). Furthermore, nilotinib demonstrated a low RMSD value throughout the MD simulation, with its RMSD generally <0.9 nm for all instances (Figure 3), with hydrophobic interactions of the main carbon chain and the amino acids noted (Figure 4). Overall, nilotinib was chosen due to the evidence in silico and due to its relatively longer half-life than other drugs (Table 2).

#### 2.3.3. Repurposable Drugs for D13L

Simeprevir was the drug chosen (Table 1). It had a high binding energy value of −9.2 kcal/mol to the D13L trimer complex, which is higher than the score of rifampin, the model drug (Figure 2). Simeprevir is previously used as an antiviral (Table 2) by inhibiting protein synthesis and, therefore, viral maturation and replication. Hydrogen bonds were noted with existing hydrogen bonds formed by the X-ray-derived rifampin (Figure 5). RMSD also showed less fluctuation than that of rifampin (Figure 3), showing strong binding stability and, thus, its potential to be a more potent inhibitor of D13L.

#### 2.3.4. Repurposable Drugs for F13L

Hypericin and naldemedine are the chosen drugs for F13L (Table 1), with their binding affinity found at −11.2 kcal/mol and −10.1 kcal/mol, respectively, both of which are higher than that of the control drug tecovirimat at −8.4 kcal/mol (Figure 2). Hypericin showed a particularly interesting pose. Having a honeycomb-like shape, it fits snugly into the F13L pocket, with the hydrophilic ends interacting accordingly with the protein (Figure 2). Two hydrogen bonds were also noted between hypericin and the neighbouring amino acid residues (Figure 4). Naldemedine was also noted to have a combination of both hydrophobic and hydrophilic areas, with the hydrophilic nitrogen group binding appropriately. In the simulation, both drugs showed consistent RMSD values of <0.5 nm (Figure 3), and clusters of the most stable conformations made up >95% of the entire simulation length, further highlighting the stability of hypericin and naldemedine in the simulation (Table 2). Multiple non-transient hydrogen bonds were also observed for naldemedine (Figure 5), further emphasising the in silico stability of the structure. Naldemedine also showed many hydrophobic interactions with the neighbouring F13L protein residues (Figure 4).

#### 2.3.5. Repurposable Drugs for I7L

Fosdagrocorat and lixivaptan (Table 1) were both identified to have the strongest binding affinity among the drugs screened at −10.1 kcal/mol compared with that of the control TTP6171, at −9.3 kcal/mol. Both fosdagrocorat and lixivaptan showed increased stability to the binding site, with much lower RMSD values and low fluctuation in the RMSD throughout the simulation (Figure 2). A strong hydrogen bond was also observed for fosdagrocorat throughout the simulation. The presence of aromatic rings in both fosdagrocorat and lixivaptan also allows for pi-stacking, further increasing factors that stabilise the ligand position in protein (Figure 2).

### 2.4. Overall Docking Affinities of All Screened Drugs

Overall docking showed that ligands fitted generally within 0.3 nm of the active site for all targets, though with varying binding affinities (Figure 2). Outside of mitoxantrone, all other established drugs had a stronger binding affinity than the average of the drug library (Figure 2). All binding energy values of the docked drugs are available in Appendix A.

### 2.5. Readjustment of Poses in Simulation

TTP-6171 as a control for I7L was found to have a higher RMSD value but stabilised more at a later stage of the simulation due to the drug changing to a different extended conformation which still binds near to the active site residues. However, its RMSD every 2.5 ns exceeded the clustering threshold of 0.3 nm and, as a result, was not binned into the > 1 cluster sets (Table 2).

### 2.6. Hydrophobic Interactions in Most Stable Conformation Identified through Clustering

For A48R, eight and nine hydrophobic interactions for NMCT and rutaecarpine were identified. For A50R, 7 were identified for the control mitoxantrone, and 10 for nilotinib. For D13L, nine were found for the control rifampin and five for simeprevir. For F13L, 5, 6, and 12 hydrophobic interactions were found for tecovirimat (control), hypericin, and naldemedine, respectively. Lastly, seven, six, and seven hydrophobic interactions were found for I7L for the control drug TTP6171, fosdagrocorat, and lixivaptan. The high number of hydrophobic interactions generally corresponded to a lower RMSD value, as derived from MD simulations.

### 2.7. Hydrogen Bonding for Experimentally Derived Structures

Hydrogen bonds were observed between ligand and receptors throughout the simulation (Figure 5). In A48R, NMCT formed a consistent hydrogen bond with Asp-13 and Arg-72. Rutaecarpine showed transient hydrogen bond formation. In D13L, consistent hydrogen bonding between Gln-27 was observed for both the control rifampin and simeprevir, indicating a potentially active and important site.

### 2.8. Hydrogen Bonding for AlphaFold2-Derived Structures

Many residues overlap in hydrogen bond formation with the ligand in A50R, F13L, and I7L. A50R displayed inconsistent hydrogen bonds throughout the simulation for all the drugs but strong hydrogen bonding for Lys-157 in mitoxantrone and Arg-8 in nilotinib. F13L showed a potential hotspot residue of Arg-89 and Ser-135, binding all the three drugs investigated. Furthermore, for F13L, all drug targets showed the formation of hydrogen bonds in their most stable conformations identified through clustering, with hypericin forming two separate hydrogen bonds with structurally neighbouring residues Ser-327 and Asn-329. For I7L, TTP-6171, the control drug, was observed to form hydrogen bonds with the greatest number of residues. However, TTP-6171 and lixivaptan did not form stable hydrogen bonds with any of the residues. Only fosdagrocorat was observed to form stable hydrogen bonds with residues Asp-258 and Gly-261, with two hydrogen bonds to each residue in total for its most stable conformation (Figure 4).

### 2.9. Overall Consistency of Non-Bonded Interactions to Previously Elucidated Active Residues in the Literature

For A48R, hydrogen bonding was observed for both NMCT and rutaecarpine to Asn-65 and Lys-105 throughout the simulation. In addition, an additional amino acid of interest, Tyr-144 (Table 3), was identified, as both NMCT and rutaecarpine showed an affinity for it throughout the simulation in the form of hydrogen bonds (Figure 5). The absence of hydrophobic interactions in the most consistent cluster is consistent with the findings reported for previously identified drugs in the literature. For A50R, Cys-11 showed hydrophobic interactions with the ligands identified. Hydrogen bonding was also observed for mitoxantrone at Cys-11 but not for nilotinib. However, strong hydrogen bonding was observed in neighbouring amino acids, namely Lys-9, Tyr-15, Asn-143, and Asn-146 (Table 3). Hydrophobic interactions in D13L are consistent with the findings reported in the literature indicating Phe-486 to be an active site for the control drug rifampin; simeprevir also showed hydrophobic interactions with these residues. Moreover, these interactions were also found in the native X-ray structure, before the simulation (Appendix A). F13L showed a very high number of hydrogen bonding in general (Figure 5), with tecovirimat, the control drug, forming hydrogen bonds with Asp-280 (Table 3). However, hypericin and naldemedine did not form hydrogen bonds throughout the 100 ns simulation with Asp-280. New residues of interest, Ser-135, Asn-329, Asp-331, and especially Asn-312 were identified from the hydrogen bonding (Table 3). Asn-312 was observed to form hydrogen bonding with all the drugs tested, including tecovirimat. Moreover, in the most stable conformation, Asn-312 formed hydrophobic interactions with both hypericin and naldemedine. The results clearly show that hydrogen bonds are pivotal to stabilising the structure, as these structures merely vibrated in place in the simulation (with the exception of tecovirimat, which flipped). Lastly, for I7L, hydrogen bonds were also noted at His-241 and the neighbouring residues Ser-240 and Lys-243. His-241 also showed hydrophobic interactions with fosdagrocorat and lixivaptan (Table 3).

### 2.10. Screening of Other Molecules in Addition to Those Listed

In addition to the molecules listed, molecular screening and MD simulations were also performed on the AlphaFold2-derived structure of protease G1L. The molecules screened for these targets dislodged during the MD. Other drugs that dislodged during the MD, or otherwise showed unstable conformations were cinacalcet for A48R, ponatinib and vitamin D for A50R, rifaximin, glecaprevir, and dutasteride for D13L; lifitegrast for F13L; baloxavir, laniquidar, nilotinib, DB06925, DB04727 and DB11913 for I7L; TMC-647055, cepharanthine, fda_1667, and fda_1348 for G1L. These drugs showed RMSD values of >1 consistently throughout the simulation, with all the tested drugs dislodging from the pocket of interest and, therefore, determined to have a limited binding ability to G1L. All binding energy values, drug SMILES, and other IDs of these screened drugs can be found in Appendix A.

## 3. Discussion

### 3.1. Amino Acid Polymorphisms

Overall, a large number of polymorphisms, premature stop codons, and duplications were observed in the French isolate for all the proteins investigated. The positions of these polymorphisms are not on any key active residues and are generally conserved, despite the broken frames. There is no reported additional virulence on the French isolate to date, and given the low possibility of a large number of protein deletions and mutations relative to other isolates gathered during the same time, the sequencing depth may be insufficient; thus, further sequencing and exploration of this isolate are required.

### 3.2. Potential of Screened Drugs In Silico

All the tested drugs showed high stability in MD simulations with the CHARMM36m force field (Figure 3). This was further validated by hydrophobic contacts (Figure 4), indicating their higher potential as a monkeypox treatment option. The results also demonstrate that the number of hydrophobic contacts is a potential contributing factor to drug stability.

### 3.3. In Silico Confirmation of A50R, F13L, and I7L Binding Sites

The stability analysis and clustered poses (Figure 3 and Appendix A) confirmed the structurally active and binding sites for AlphaFold2-derived proteins. Mitoxantrone and tecovirimat were shown to dock strongly and stably to the pockets near the active sites in the entire simulation. This is in silico evidence for the location of the pocket, and further validation of previous work that proved the efficacy of the abovementioned drugs in inhibiting poxviruses [19]. TTP-6171 also showed relatively good binding to the I7L pocket, further proving its ability to stop viral replication [31] and further making its case for being a potential drug to treat monkeypox.

### 3.4. Elucidation of Potentially New Residues of Interest

For A48R, a new amino acid of interest, Tyr-144, was identified. Gln-27 for D13L also showed hydrogen bonding to both rifampin and simeprevir, highlighting its potential importance for future studies. Ser-135, Asn-329, Asp-331, and Asn-312 were the other residues noted of importance for F13L (Figure 4 and Figure 5). The importance of these residues may further facilitate the drug discovery process for other poxviruses and serve as a watch point for potential antiviral resistance mutations.

### 3.5. Half-Life, Contraindications, Side Effects, and Bioavailability of Drugs

NMCT (for target A48R) is a potent antiviral that has not been clinically used and is not currently approved but has potential potent antiviral activity in HSV-2-infected mice [42], with little apparent cytotoxicity in uninfected Vero cells [43]. In addition to being demonstrated in this study in silico for being effective in blocking A48R, NMCT also has known effects in halting viral replication by targeting J2R, a thymidine kinase in vaccinia [39]. This property of NMCT makes it a possible double-action drug in the fight against monkeypox.

Rutaecarpine (for target A48R) is a COX-2 inhibitor and is touted as an anti-inflammatory drug [44]. It is isolated in a traditional Chinese medicine “Wu-Chu-Yu” and has officially been incorporated into the Chinese Pharmacopoeia 2015 Edition [35]. It is, however, not approved by the FDA, and the half-life is unknown. The fruit in which it is found also contains other alkaloids as part of the Rutaceae family.

Nilotinib (for target A50R) is an orally administered, FDA-approved drug for the treatment of chronic myelogenous leukaemia (CML). It has an adverse drug effect in patients with a polymorphism in UGT1A1. Otherwise, it has been demonstrated to be safe in human trials [36].

Simeprevir (for target D13L) binds extensively to plasma proteins, mainly albumin [36]. Simeprevir has previous use in treating HCV, and its high oral bioavailability of 44% makes it a potentially good target [45]. It is also to be noted that the use of simeprevir is contraindicated in those with asymptomatic hepatitis B virus (HBV) infection, as usage can lead to the reactivation of the virus and subsequent symptomatic disease [46].

Hypericin (for target F13L) is a naturally occurring substance found in St. John’s Wort that can also be separately synthesised [47]. It can induce photocytotoxicity [48] and has been touted as a potential treatment and detection of cancer [47]. The photosensitising effects of hypericin have also shown its efficacy as a multimodal antiviral [49], and it has long been known to inactivate murine cytomegalovirus (MCCV) and HIV-1, especially with exposure to fluorescent light [50]. However, the use of St. John’s Wort, by extension hypericin, in combination with other medicines, has previously proven an antagonistic effect [51] and could interfere with contraceptives [52]. Otherwise, it is well-tolerated [53].

Naldemedine (for target F13L) is a peripherally acting μ-opioid receptor antagonist for opioid-induced diarrhoea in adults with chronic pain [54]. It is generally well-tolerated, even in the long term, and does not cause opioid withdrawal [55]. A previous phase III trial has demonstrated it to be safe and effective [56]. However, it has a relatively short half-life of 11 h, and constant administration may be required. Side effects include headache, nausea, and bowel dysfunction [54].

Fosdagrocorat (for target I7L) was reported to not be systemically absorbed after oral administration [57]. Therefore, IV administration may be used as a possible alternative solution for drug uptake.

Lixivaptan (for target I7L) is an orally active, non-peptide, competitive, and selective antagonist for the vasopressin V2 receptor which functions by promoting the excretion of retained body water [58]. The half-life of lixivaptan is approximately 11 h [38]. The longer half-life allows the drug to be present for a longer duration and exhibit its suitability as a potential I7L inhibitor.

### 3.6. Antiviral Resistance towards Derived Drugs

Rifampin, a previously identified drug that binds to D13L, has previously not been used as an antiviral in treating poxvirus infections due to its development of antiviral resistance [24]. Moreover, the presence of a strong Gly-27 bond between D13L and both simeprevir and rifampin indicates a possible hotspot for antiviral resistance, as a mutation of this molecule may signal a loss of a strong, consistent hydrogen bond. In other sites, previous mutations in vaccinia have also shown resistance to mitoxantrone and tecovirimat [19,31]. It is through these sites that the active residues were identified; however, these mutations may also equally render other drugs useless. In order for this to be overcome, a drug cocktail may be used, such as a drug prescribed in highly antimicrobial resistant settings [59], to completely wipe out the virus without giving it mutation potential under evolutionary stress.

### 3.7. Limitations of the Study

The authors acknowledge the following four limiting factors (LF) of this study:

**LF1.** The bioavailability of some drugs is not fully known. Since all the investigated targets are found in the cytoplasm of an infected cell, a lack of plasma membrane permeability could in reduced efficacy. This would make drug titration difficult. Furthermore, some drugs may be metabolised differently in vivo, rendering the metabolite not effective.

**LF2.** Docking and simulations are purely theoretical tools. Due to the imperfectness of the energy functions used, false-positive cases could not be avoided.

**LF3.** The prediction of A50R, F13L, and I7L structures was performed using AlphaFold2. The accuracy of these predictions as a result could not be fully validated.

**LF4.** The true efficacy of the repurposed drug could not be demonstrated in silico. More evidence, especially in vivo studies, is required to ascertain the drug’s usability for monkeypox treatment before it is widely applied.

## 4. Methods

### 4.1. Multiple Sequence Alignment with Other Poxviruses

Poxvirus sequences were obtained from the NCBI Nucleotide database (GenBank and EMBL accessions: ON585029.1-monkeypox, 2022 Portuguese isolate; ON568298.1-monkeypox, 2022 German isolate; ON602722.1-monkeypox, 2022 French isolate; and LT706528.1-smallpox, historical sample). Proteins of interest were identified using tBLASTn 2.12.0+, translated using the ExPASY [60] translate tool, and subsequently aligned using MultAlin [61].

### 4.2. Structures Used for Virtual Screening and Molecular Dynamics (MD)

A48R and D13L structures were modelled using previously experimentally determined structures, PDB 2V54 and 6BED, respectively. Additional hydrogen atoms were not augmented onto the experimentally determined proteins. A50R, F13L, and I7L were modelled using AlphaFold2 [62] with a ColabFold wrapper [63]. The protein sequences used were from the German isolate of monkeypox, extracted using tBLASTn 2.12.0+ (NCBI accessions—A50R: AAO89455.1, F13L: AAO89331.1, I7L: AAO89355.1). The extracted sequences were translated using the ExPASY [60] translate tool before AlphaFold2 structural prediction.

### 4.3. Docking Site Identification from Structures

The active sites for A48R and D13L were identified from the location of thymidine and rifampin in their respective PDB structures. A50R, F13L, and I7L active sites were identified from mutation studies in the literature [21,31,41] and mapped to the monkeypox analogues after multiple sequence alignment.

### 4.4. Reverse Docking for Pocket Validation for AlphaFold2 Predicted Proteins

Reverse docking was performed using mitoxantrone for A50R, tecovirimat for F13L, and TTP6171 for I7L, and visual screening was performed to validate the correct site.

### 4.5. Molecular Screening and Docking

QuickVina 2.1 (QVina2.1) [64] was used in conjunction with the ZINC library [65] of FDA drugs and drugs approved by other major jurisdictions (non-FDA), and also DrugBank [36,66]. The partial charges of ligand molecules were calculated using ADFR Tools [67] using the default Gasteiger charges. The sorting of the top candidates was performed with the top performer having the largest binding affinity score to the site of interest. The manual inspection of docking sites and docking poses were then performed using PyMOL 2.5.0 [68]. PDBQT files from QVina2.1 were converted back into PDB and appended for CHARMM-GUI using OpenBabel 3.1.0 [69]. Docking was performed using mitoxantrone for A50R, tecovirimat for F13L, and TTP6171 for I7L, and a visual inspection was performed to validate the correct binding site.

### 4.6. Molecular Dynamics Simulations

MD simulations and energy minimisation were performed using GROMACS version 5.1.8 [70]. The CHARMM36m force field was used and added using CHARMM-GUI’s solution builder [71,72] after a separate ligand force field was constructed. GalaxyFill [73] was used as per CHARMM-GUI to fill in gaps present in the X-ray structure of A48R and D13L so that MD could be performed. A cubic periodic boundary condition (PBC) of 1 nm at the farthest end of each protein boundary was established. The protein was solvated in a TIP3P water solvent at 0.15 M of KCl, and energy minimisation was performed using the steepest integrator at 0.1 fs for 10,000 iterations, using the potential of mean force (PME) for coulomb, non-bonded minimisation. Following this, equilibration was performed at 1 fs for 125,000 iterations, using the Verlet integrator, equilibrating the system to 303.15 K. Lastly, a total NPT ensemble simulation of 100 ns was performed at 2 fs per iteration. V-rescale and Parrinello–Rahman were used for temperature (303.15 K) and pressure coupling, respectively (1 bar). Position restraints were not used in either step. Covalent hydrogen bonds were constrained in the equilibration and MD steps.

### 4.7. Assessment of Protein Stability

Ramachandran plots were used to assess the stability of both AlphaFold2-predicted and experimentally derived protein structures and were plotted using the Ramachandran plot server [74]. Furthermore, root mean square fluctuation (RMSF) and root mean square distance (RMSD) were calculated to assess stability.

### 4.8. Calculation of MD Metrics

RMSD was calculated of the ligand to its initial structure. The minimum distance between the ligand and the receptor and hydrogen bonds were also computed. Hydrogen bond interactions in each ligand–receptor were also captured with a cutoff radius of 0.35 nm and cutoff angle of 30 degrees. All metrics were calculated at a timestep of 1 ps and performed using GROMACS. Both RMSD and minimum distance were computed after rotation and fitting to the initial structure. The RMSF and RMSD values of proteins were calculated relative to the initial structure of the protein for all MDs with ligands after centring, adjusting for PBC, and fitting to the pocket of interest.

### 4.9. Clustering and Ligand–Receptor Interaction Classification

Clustering was performed using the GROMOS method [75] with an RMSD threshold of 0.3 nm per cluster on a simplified trajectory at 2.5 ns per frame. Ligand plots and associated interactions were determined using LIGPLOT+ (version 2.2.5, 27 January 2022) [76] for the cluster with the highest number of frames.

## 5. Conclusions

With zoonotic diseases being of increasing threat to human survivability—and monkeypox amongst other poxviruses becoming an emerging threat—it is imperative for a greater exploration of pharmacopoeia to address these diseases. The use of potential targets NMCT and rutaecarpine for A48R, nilotinib for A50R, simeprevir for D13L, hypericin and naldemedine for F13L, fosdagrocorat and lixivaptan for I7L, and the controls named in this paper should be further investigated so as to reduce fatalities caused by viruses. The authors also hope that the uprise of recent pandemics and epidemics, namely COVID-19 and monkeypox, will garner more efforts towards focusing much-needed attention on diseases, currently neglected or otherwise, and prompt much-warranted research.

## Figures and Tables

**Figure 1 molecules-27-05277-f001:**
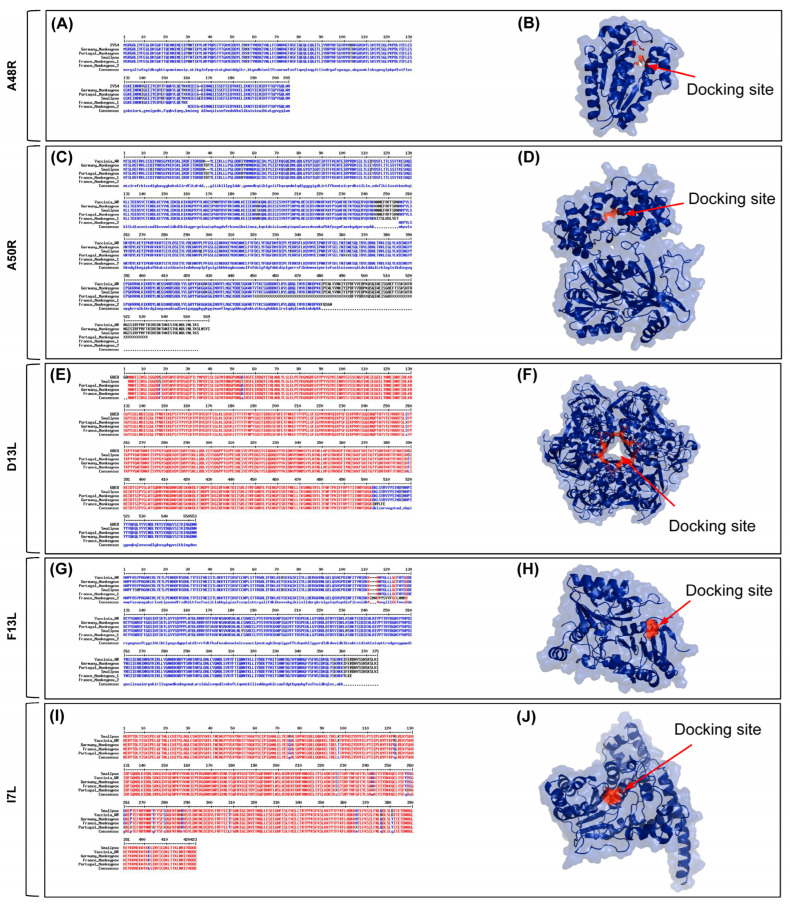
Significant similarity of (**A**) A48R 2V54, (**C**) A50R, (**E**) D13L 6BED, (**G**) F13L, and (**I**) I7L across vaccinia, smallpox, and monkeypox. Sequences were aligned using MultAlin. Protein structures: (**B**) PDB: 2V54 from vaccinia, (**D**) AlphaFold2-predicted structure for German isolate A50R protein, (**F**) PDB: 6BED from vaccinia, (**H**) AlphaFold2-predicted structure for German isolate F13L protein, and (**J**) AlphaFold2-predicted structure for German isolate I7L protein.

**Figure 2 molecules-27-05277-f002:**
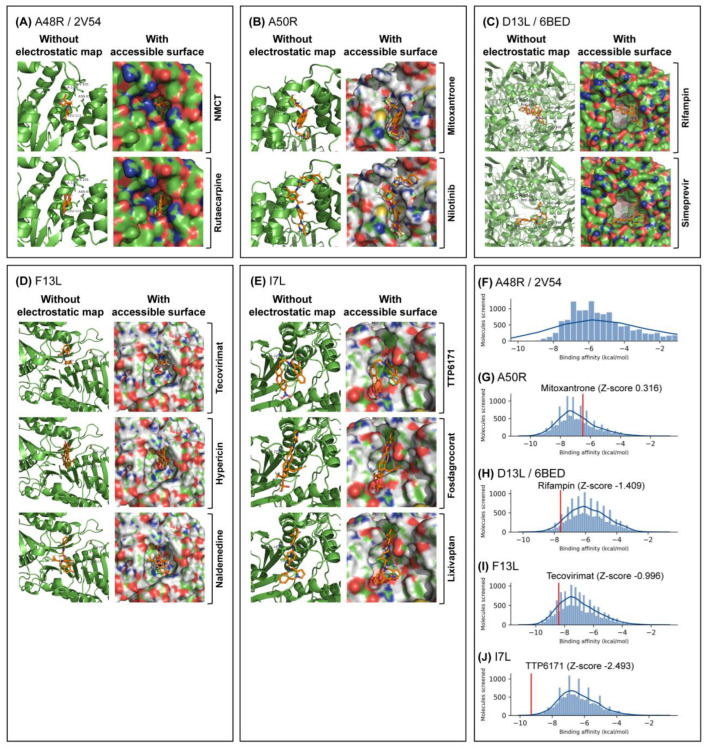
Identified drugs of interest show good fit within docking regions and pockets, and control drugs generally scored higher than the majority of docked drugs, with the exception of mitoxant. rone in A50R. Docking was performed using QVina2.1 around the region of interest, with a search box of 30 × 30 angstroms, with the best poses sorted by strongest binding affinity shown for (**A**) A48R, (**B**) A50R, (**C**) D13L, (**D**) F13L and (**E**) I7L. Electrostatic maps with solvent-accessible surfaces were calculated using PyMOL, with individual amino acids of importance labelled on diagrams. Distribution of QVina2.1 binding affinities are shown for (**F**) A48R, (**G**) A50R, (**H**) D13L, (**I**) F13L, (**J**) I7L, with the control drugs and binding affinities indicated in red.

**Figure 3 molecules-27-05277-f003:**
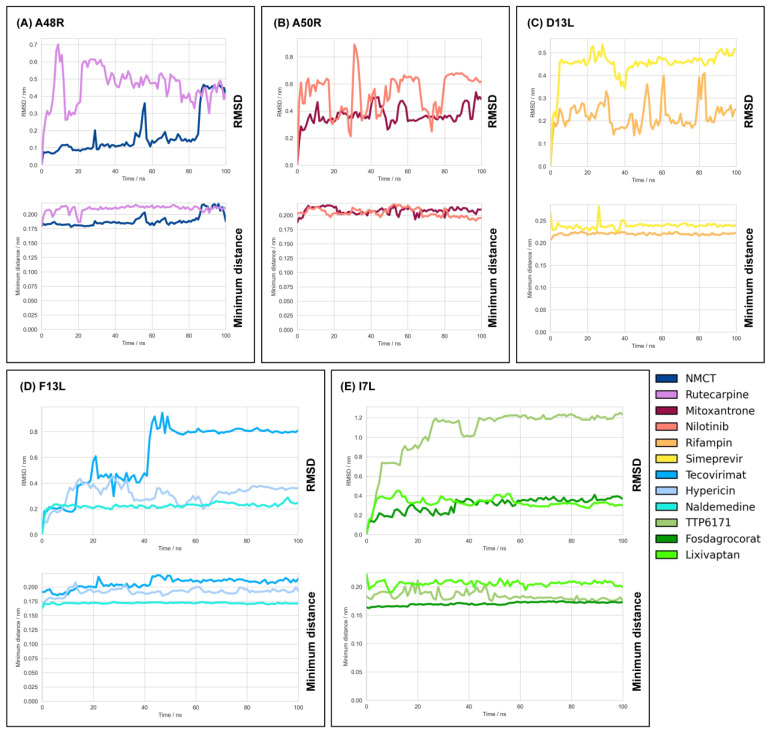
High stability of drugs observed in MD simulations, with all the proposed novel drugs having an RMSD < 1 and control drugs having an RMSD < 1.3. Molecular dynamics were performed using the CHARMM36m force field on (**A**) A48R, (**B**) A50R, (**C**) D13L, (**D**) F13L and (**E**) I7L. RMSD of ligand was computed using amino acids 1 to 141 for (**B**) A50R, chain C for (**C**) D13L, and computed using whole protein for the others. Moving averages of 1 ns were plotted, respectively.

**Figure 4 molecules-27-05277-f004:**
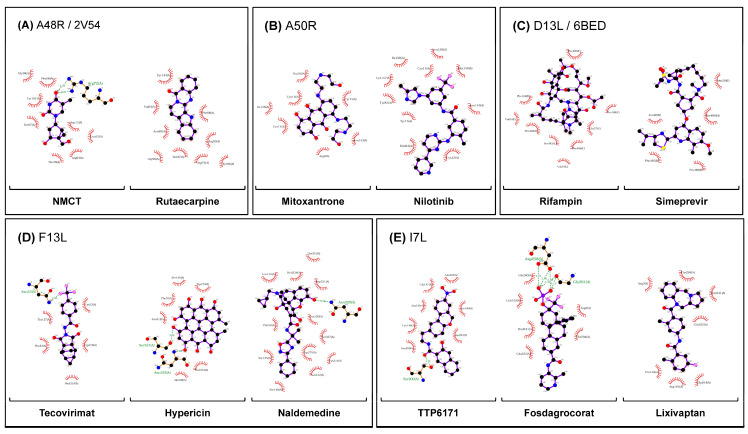
Strong non-bonding interactions observed between most clustered poses of ligands. (**A**) A48R, (**B**) A50R, (**C**) D13L, (**D**) F13L and (**E**) I7L trajectories were clustered using GROMOS force field at an RMSD cutoff of 0.3 nm in GROMACS, with the cluster with highest number of trajectory timesteps analysed. Ions and water molecules were removed before analysis, as their interactions are transient. LigPLOT+ was used to generate the ligand interaction figures.

**Figure 5 molecules-27-05277-f005:**
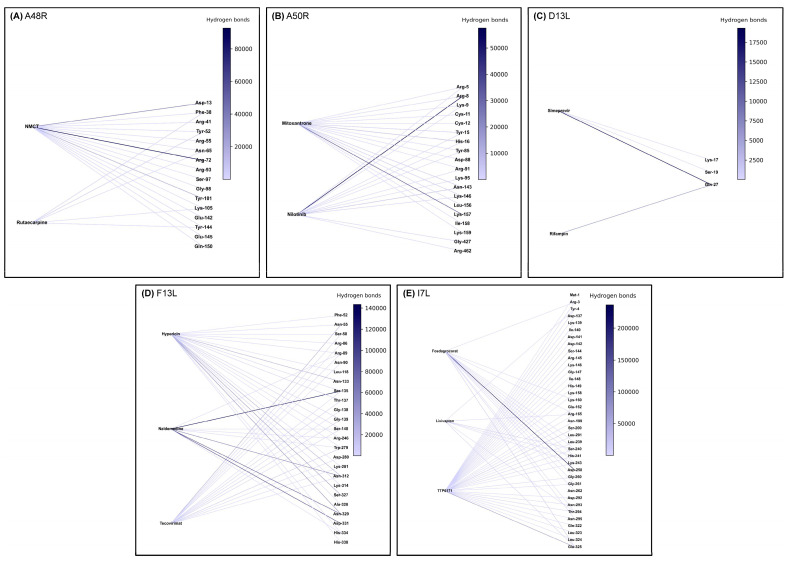
Hydrogen bonds to potential residues of interest. Hydrogen bonds were calculated with a cutoff of 0.3 nm and 30 degrees in GROMACS for the whole 100 ns simulation for (**A**) A48R, (**B**) A50R, (**C**) D13L, (**D**) F13L and (**E**) I7L. Colour intensity indicates the number of hydrogen bonds to a specific residue summed throughout the simulation.

**Table 1 molecules-27-05277-t001:** Potential drugs derived for monkeypox targets.

Gene	Protein Function	Reference Protein	In Vitro Proven Drug	Novel Repurposable Drug	DrugStructure
A48R	Thymidylate kinase	PDB: 2V54 [20]	-	NMCT	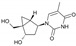
Rutaecarpine	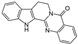
A50R	DNA ligase [19]	AlphaFold2 structure of German monkeypox viral protein identified via tBLASTn	Mitoxantrone [19,21]	Nilotinib	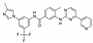
D13L	Viral capsid protein [19]	PDB: 6BED [24]	Rifampin [19]	Simeprevir	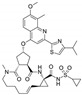
F13L	EEV formation protein [19]	AlphaFold2 structure of German monkeypox viral protein identified via tBLASTn	Tecovirimat [19]	Hypericin	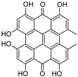
Naldemedine	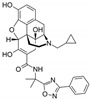
I7L	Protease [19]	AlphaFold2 structure of German monkeypox viral protein identified via tBLASTn	TTP6171 [19]	Fosdagrocorat	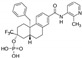
Lixivaptan	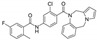

**Table 2 molecules-27-05277-t002:** Drug binding stability in simulation and pharmacology.

Target	Drug	Largest Cluster Middle RMSD/nm	Top Cluster Proportion/%	Drug Half-Life	Current Drug Uses
A48R	NMCT	0.172	100.00	3.7 h (in vitro cells only)	Nucleoside analogue, potent viral inhibitor, not used in humans [34]
	Rutaecarpine	0.173	100.00	Unknown	COX-2 inhibitor for headache and abdominal pain in traditional Chinese medicine [35]
A50R	Mitoxantrone (control)	0.217	90.24	75 h [36]	Multiple sclerosis [36]
	Nilotinib	0.238	29.27	15 h [36]	Leukaemia [36]
D13L	Rifampin/rifampicin (control)	0.175	100.00	3.35 (±0.66) h [36]	Antibiotic [36]
	Simeprevir	0.193	90.24	10–13 h [36]	Antiviral [36]
F13L	Tecovirimat (control)	0.222	100.00	21 h (intravenous), 19 h (oral) [36]	Poxvirus treatment [36]
	Hypericin	0.194	97.56	24.8–26.5 h [37]	Supplement, part of St. John’s Wort
	Naldemedine	0.219	100.00	11 h [36]	Opioid-induced constipation [36]
I7L	TTP6171 (control)	-	-	Unknown	Protease inhibitor, not used in humans
	Fosdagrocorat	0.214	56.10	Unknown	Rheumatoid arthritis [36]
	Lixivaptan	0.256	58.54	11 h [38]	Hyponatremia, congestive heart failure [36]

**Table 3 molecules-27-05277-t003:** Active residues in identified targets and potentially new active residues of interest.

Gene	Active Residues (Per the Literature) Relative to Reference Protein	Potentially New Active Residues of Interest (Per Results) Relative to Reference Protein
A48R	Asn-65, Leu-111, Lys-105, Ala-107 (structural study) [20]	Tyr-144
A50R	Cys-11, Cys-147 (mutation study) [21]	Lys-9, Tyr-15, Asn-143, Asn-146
D13L	Phe-168, Phe-486, Phe-487 (mutation study) [40]	-
F13L	Gly-277, Asp-280 (mutation study) [41]	Ser-135, Asp-280,, Asn-329, Asp-331
I7L	His-241, Cys-328 (mutation study) [31]	Ser-240, Lys-243

## Data Availability

All QVina2.1 binding affinities for the screened drugs are provided in Appendix A. Raw molecular dynamics trajectories will be provided by the authors upon request. It is worth noting that trajectories are > 50 GB each, with all full files (including rotated, centred trajectories, etc.) > 300 GB per MD. The transfer will be at the expense of the requester. Summarised molecular dynamics trajectories are provided in Appendix A.

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
