# Peer review of "In Silico Repurposed Drugs against Monkeypox Virus"

_molecules, 2022, doi:10.3390/molecules27165277_

Round 1
Reviewer 1 Report
The authors report on in silico drug repurposing targeting monkeypox virus which is an emerging epidemic of concern. Virtual screening and molecular dynamics were employed to explore the potential repurposing of multiple drugs previously approved by the FDA. The results provide immediately testable hypotheses and are of value for the antiviral research community.
The manuscript is clear and can be improved by showing the 2D chemical structure descriptions of the identified compounds in Table 1 or Figure 4.
Reviewer 2 Report
The manuscript authored by Yuen In et al, performed Insilico analysis on the repurpose of FDA approved drugs against monkeypox virus. The manuscript is well organized and written. However, the minor revision is required before acceptance for the publication.
Minor comments
1) Please include the validations of the protein models generated from alpha fold, for example, Ramachandran plot in the supplemental files.
2) “A cubic periodic boundary condition (PBC) of 10 nm at the farthest end of each protein boundary was established” this sentence is unclear because,10-nm box amounts 100Å which is arguably large. Please check it.
3) The figures mentioned in the manuscript text are not in the order, for example (page 7 line 207 and 209) figure 5 is mentioned after figure 2. Please order the figures and supplemental figures in the text.
4) Though, the potentially new residues of interest for each protein is provided in the manuscript text, it would be better to provide as another table.
Reviewer 3 Report
The paper presents in silico investigation on the repurposing drugs against monkeypox virus. Despite the performed investigation and obtained results, the paper needs serious additional efforts in order to meet the scientific level to be published in such an esteemed journal as Molecules.
Here are some of the major points need to be considered before further consideration for possible publication:
1. How have been chosen the five poxvirus targets?
2. How exactly “the authors mapped the active residues to monkeypox and propose eight novel potentially repurposable drugs”?
3. I find inappropriate Section “Results” to start directly with a Table with results… In addition, Table 1 is not cited in the text.
4. I find subsections 3.2. and 3.3. totally uninformative, as presented by one sentence and redirecting to supplementary figure and supplementary file, without any analysis and discussion.
5. Citations of Figs. 5 and 4 (in this order) and the figures themselves appear before Fig. 3, which appears far, far later… Please consider renumbering of figures.
6. Why “Ions and water molecules were removed before analysis.” (legend in Fig. 4)?
7. Fig. 4 presents “Strong non-bonding interactions”, while later in the text there are statements like “Two hydrogen bonds were also noted between hypericin and the neighbouring amino acid residues (Fig. 4)”, “two hydrogen bonds to each residue in total for its most stable conformation (Fig. 4).”. Please explain the discrepancies… Moreover, Fig. 4 is with a low quality, so no much information can be obtained from it.
8. Why there is no control presented for A48R in Table 2?
9. A part of the information in the legend of Fig. 3 is more appropriate to be presented in Methods Section.
10. Concerning “3.15. Screening of other molecules in addition to those listed.” – although an additional investigation performed, it is not enough just to redirect to “All binding energies, drug SMILES, and other IDs can be 349 found in Supplementary File 1.”, without any analysis and discussion.
11. I would suggest reorganization of the results obtained target by target, but not as many subsections (now 15) in Results Section… I would also suggest a combination of Results and Discussion Sections in one, since now the presentation is too fragmented in so many subsections…
12. Since I could not find any restrictions, I would suggest incorporating the three supplementary figures into the main text. From one hand, they are not too big, from the other – it will make reading and understanding the results easier.
Reviewer 4 Report
Recently, monkeypox is a popular topic (similar to Covid-19). This work is only theoretical studies, without any experiment. Therefore, as the authors rightly noted, the manuscript has many limitations. Especially LF4. It is good that the authors are aware of that. I have some comments:
1. The order of the figures is not correct, also in the text. Figure 1, 2, then 5, 4, 3. In the text, the figures should be cited sequentially, Fig.1,2,3, 4, 5
2. The figures are not readable, especially Fig. 2 and 4. The authors briefly describe how the studied drugs interact in the pockets. And it is impossible to read it from the figures.
3. There are no figures S1, S2, S3 in the supplementary file. The supplementary file is only an excel file with many data.
4. Molecular Dynamic: RMSD is ok, but usually the fluctuations are discussed in relation to proteins without ligand. In my opinion, the authors should include in figure 3 a reference RMDS plot of peptide (gene) and gene+native ligand (just like for F13L and I7L; why for other plots there are not RMDS plots for native complex?). And the discussion should be expanded.
5 Molecular Dynamic: RMSD is a good parameter for determining the stability of the complex. But it would be also interesting to obtain the RMSF value to explore local protein flexibility. For protein and complex. And discuss it.
6. The formatting of table 1 is not correct
7. I am not sure, but formatting of references is not correct for Molecule.
Round 2
Reviewer 3 Report
The authors present a revised version of the paper where most of my suggestions had been addressed. However, I still have a few points to be clarified before proceeding further with the manuscript publication:
1. The authors honestly say that “Indeed, further elaboration is required and discussion is required for this section, especially for that related to the Supplementary Figure mentioned.” After such “confession” and following my belief as well, I would suggest removing of Supplementary Fig. 1.
2. Concerning the authors’ response about “Strong non-bonding interactions” and Fig. 4: if it is a question of the figure quality, this problem should be solved accordingly, before to proceed further.
3. In the revised version, supplementary figures appear at the end of the manuscript… I respect the authors’ opinion “… are in our view auxiliary and may serve to dilute the core message…”, and as such they should be just explained in the main text, as it was in the first version and not to appear as it is in the current version.
Reviewer 4 Report
The authors of the paper responded to my comments. I have no objections to most of them. But still, I have minor comments on figures and formatting:
2. “Response: We have previously submitted 300 DPI images to the Journal, however upon conversion to a Word document and editorialising by the Journal, we believe that the image was compressed, leading to the poor quality. We will forward this feedback to the journal.”
I don't think it's a matter of compression or resolution. Figures 2A-E and 4 are too small, more precisely, the names of the acid residues surrounding the drug, bond lengths etc. I am not able to read it
6. “Response: Appreciate the feedback. We have reformatted the table accordingly as shown below. Please let us know if you would prefer it formatted in another method.”
The width of columns 4 and 5 is too small. For example:
Mitoxantro
ne
Tecovirima
t
it should be in one line
Maybe you need to lower the case size?
7. “Response: Thank you for pointing this out. We have checked with the Journal specifications, and the Instructions for Authors page states: “Your references may be in any style, provided that you use the consistent formatting throughout. It is essential to include author(s) name(s), journal or book title, article or chapter title (where required), year of publication, volume and issue (where appropriate) and pagination. DOI numbers (Digital Object Identifier) are not mandatory but highly encouraged”.
Ok, but the formatting is not correct, for example for journals should be:
1. Author 1, A.B.; Author 2, C.D. Title of the article. Abbreviated Journal Name (italic) Year (bold), Volume (italic), page range.
See the instructions for authors or word template
In this paper for example:
Ref 15 and 17 – the same journal, once the title is normal the second time it is italic. ??
All the journal references are badly formatted.
From the page instructions for authors, you can download the output style for the endnote.
